

# Small-scale screening of anticancer drugs acting specifically on neural stem/progenitor cells derived from human-induced pluripotent stem cells using a time-course cytotoxicity test

Hayato Fukusumi[1], Yukako Handa[2], Tomoko Shofuda[1] and Yonehiro Kanemura[2,3,4]

[1] Division of Stem Cell Research, Institute for Clinical Research, Osaka National Hospital, National Hospital Organization, Osaka, Japan
[2] Division of Regenerative Medicine, Institute for Clinical Research, Osaka National Hospital, National Hospital Organization, Osaka, Japan
[3] Department of Neurosurgery, Osaka National Hospital, National Hospital Organization, Osaka, Japan
[4] Department of Physiology, Keio University School of Medicine, Tokyo, Japan

Corresponding author
Yonehiro Kanemura, kanemura@onh.go.jp

## ABSTRACT

Since the development of human-induced pluripotent stem cells (hiPSCs), various types of hiPSC-derived cells have been established for regenerative medicine and drug development. Neural stem/progenitor cells (NSPCs) derived from hiPSCs (hiPSC-NSPCs) have shown benefits for regenerative therapy of the central nervous system. However, owing to their intrinsic proliferative potential, therapies using transplanted hiPSC-NSPCs carry an inherent risk of undesired growth *in vivo*. Therefore, it is important to find cytotoxic drugs that can specifically target overproliferative transplanted hiPSC-NSPCs without damaging the intrinsic *in vivo* stem-cell system. Here, we examined the chemosensitivity of hiPSC-NSPCs and human neural tissue—derived NSPCs (hN-NSPCs) to the general anticancer drugs cisplatin, etoposide, mercaptopurine, and methotrexate. A time-course analysis of neurospheres in a microsphere array identified cisplatin and etoposide as fast-acting drugs, and mercaptopurine and methotrexate as slow-acting drugs. Notably, the slow-acting drugs were eventually cytotoxic to hiPSC-NSPCs but not to hN-NSPCs, a phenomenon not evident in the conventional endpoint assay on day 2 of treatment. Our results indicate that slow-acting drugs can distinguish hiPSC-NSPCs from hN-NSPCs and may provide an effective backup safety measure in stem-cell transplant therapies.

## INTRODUCTION

Since the development of human-induced pluripotent stem cells (hiPSCs) (*Takahashi et al., 2007*; *Yu et al., 2007*), various types of hiPSC-derived cells have been established that can be

used in regenerative medicine and drug development, while avoiding many of the ethical issues and technical difficulties involved with human tissue–derived cells. Human iPSC-derived neural stem/progenitor cells (hiPSC-NSPCs) (*Fujimoto et al., 2012*; *Kobayashi et al., 2012*; *Oki et al., 2012*; *Tornero et al., 2013*) and human fetal neural tissue–derived NSPCs (hN-NSPCs) (*Ishibashi et al., 2004*; *Iwanami et al., 2005*; *Ogawa et al., 2002*) have proven beneficial in treating various central nervous system diseases and injuries. However, the intrinsic proliferative potential of hiPSC-NSPCs, which makes them promising sources for large numbers of cells *in vitro*, can be a double-edged sword *in vivo*: transplanted cells can proliferate excessively before terminal differentiation in specific microenvironments. Although such undesired proliferation has not generally produced teratomas, malignant carcinogenesis, or other serious adverse events (*Nori et al., 2015*; *Sugai et al., 2016*), this inherent potential suggests the need for backup safety measures for stem cell–based therapies.

One strategy for reducing the risk of overgrowth is to transduce a gene that can induce apoptosis, such as herpes simplex virus truncated thymidine kinase (HSV-tk) activated by ganciclovir (*Cao et al., 2007*) or a caspase-based artificial cell-death switch (iCaspase-9) activated by AP20187 (*Krishnamurthy et al., 2010*), into the stem-cell genome. However, inserting exogenous genes into the donor-cell genome contradicts the purpose of integration-free hiPSCs, which are generated to minimize the risk of genetic modification or transgene re-activation, and transgenic strategies may create new risks despite the use of 'genomic safe harbors' for insertions in the human genome. Another strategy is to use drugs to suppress the *in vivo* overgrowth of transplanted cells; for instance, pretreating hiPSC-NSPCs with a γ-secretase inhibitor inhibits Notch signaling, which is required for maintaining NSPC stemness (*Okubo et al., 2016*). However, a single treatment prior to transplantation may not be sufficient to overcome the cells' growth potential, and cannot regulate cell growth after transplantation. Therefore, a useful backup safety measure would be a method to chemically ablate transplanted cells, preferably with a cytotoxic drug that specifically acts on transplanted hiPSC-NSPCs but not tissue-resident NSPCs.

In this study, we assessed four approved anticancer drugs, two cytotoxic (cisplatin and etoposide) and two cytostatic (mercaptopurine and methotrexate), as candidates for suppressing the overgrowth of non-transgenic stem cells *in vivo*.

Although the efficacy of candidate drugs has conventionally been evaluated by cell-destructive methods, such as MTT or ATP assays, these methods cannot assess the effects of a drug on the same cell population over time. To address this, previous studies have assessed the time-course of pharmacological effects using cell-nondestructive methods, such as measurement of changes in impedance in two-dimensional (2D) adherent cell cultures (*Caviglia et al., 2015*) and image-based measurement of the spheroid size in three-dimensional (3D) cultures of various cell types, including glioma cells (*Vinci et al., 2012*), hepatocytes (*Bell et al., 2016*), and cardiomyocytes (*Beauchamp et al., 2015*). It is known that 3D culture systems mimic the *in vivo* environment more effectively than 2D culture systems (*Achilli, Meyer & Morgan, 2012*; *Pampaloni, Reynaud & Stelzer, 2007*). Therefore, the present study used a conventional endpoint assay on day 2 of the treatment and a 7-day time-course cytotoxicity test to determine the effects of cisplatin, etoposide,

mercaptopurine, and methotrexate on 3D neurospheres derived from hiPSC-NSPCs and hN-NSPCs, which are considered to mimic the *in vivo* stem cell system.

## MATERIALS AND METHODS

### Ethics statement

This study was conducted in accordance with the principles of the Declaration of Helsinki. The use of hN-NSPCs and hiPSCs was approved by the Osaka National Hospital hN-NSPCs and hiPSCs ethics committee (Nos. 110, 120, and 146).

### Cell lines

We used two hN-NSPC lines (oh-NSC-3-fb and oh-NSC-7-fb) (*Kanemura et al., 2002*) and two hiPSC (201B7)-derived NSPC lines: the DSM line, which was established using the single SMAD-inhibition method with the Noggin alternative dorsomorphin (DSM) (*Shofuda et al., 2013*), and the dSMAD line, which was established by the dual SMAD-inhibition method with DSM and SB431542 (*Fukusumi et al., 2016*).

### Cell culture

The hN-NSPCs and hiPSC-NSPCs were propagated as neurospheres in Dulbecco's Modified Eagle's Medium (DMEM)/F12 (D8062; Sigma-Aldrich, St. Louis, MO, USA) with 15 mM HEPES (Sigma-Aldrich), epidermal growth factor (EGF, 20 ng/mL; PeproTech, Rocky Hill, NJ, USA), fibroblast growth factor 2 (FGF2, 20 ng/mL; PeproTech), leukemia inhibitory factor (LIF, 10 ng/mL; Millipore, Billerica, MA, USA), B27 supplement (B27, 2%; Thermo Fisher Scientific, Waltham, MA, USA), and heparin (5 μg/mL; Sigma-Aldrich) (*Fukusumi et al., 2016*; *Kanemura et al., 2002*; *Shofuda et al., 2013*). For hN-NSPCs, half of the medium was changed once a week. The neurospheres were dissociated into single cells every 14 days by incubating them with 0.05% trypsin/EDTA (Thermo Fisher Scientific) at 37 °C for 20 min, after which soybean trypsin inhibitor (Roche, Basel, Switzerland) was added to stop the enzyme activity. The cells were then resuspended in 50% fresh medium plus 50% conditioned medium at a density of $1 \times 10^5$ cells/mL (*Kanemura et al., 2002*). For hiPSC-NSPCs, the medium was changed every 3–5 days. The cells were passaged every 10–12 days using Accutase (Innovative Cell Technologies, San Diego, CA, USA) at 37 °C for 10 min for single-cell dissociation, after which the cells were resuspended in 100% fresh medium at a density of $1 \times 10^5$ cells/mL (*Fukusumi et al., 2016*; *Shofuda et al., 2013*).

### Drug preparation

Cisplatin (Sigma-Aldrich), etoposide (Sigma-Aldrich), mercaptopurine (Sigma-Aldrich), and methotrexate (LKT Laboratories, St. Paul, MN, USA) were dissolved in dimethyl sulfoxide (DMSO) to generate 100 mM stock solutions.

### Endpoint (ATP) assay

The neurospheres were dissociated into single cells and seeded into 96-well plates at a density of $3 \times 10^4$ cells/well (day −1). On day 0, cisplatin, etoposide, and methotrexate were applied at 0, 0.1, 0.3, 1, 3, 10, 30, and 100 μM, and mercaptopurine was applied at 0, 1, 3, 10, 30, 100, 300, and 1,000 μM (0 μM indicates DMSO only). ATP content was assayed

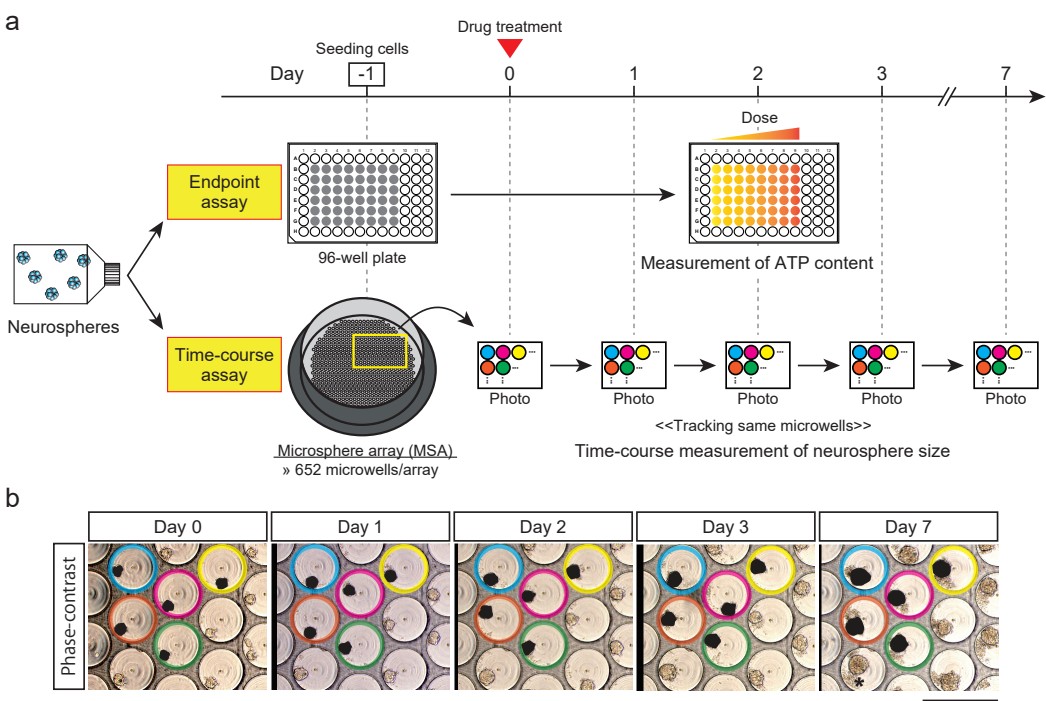

**Figure 1 Drug-screening strategies.** (A) Schematic of endpoint assay (ATP assay) and time-course cytotoxicity test. Cells were cultured in a standard 96-well plate for the endpoint assay and in a microsphere array (MSA) for the time-course cytotoxicity test. Same-colored circles indicate the same microwells in the panels for days 0, 1, 2, 3, and 7. (B) Representative phase-contrast images taken on days 0, 1, 2, 3, and 7. One field contains approximately 31 microwells. Same-colored circles indicate the same microwells during the test. Black areas are the estimated areas of neurospheres, which consist of viable cells. The * on the day-7 panel shows the locations of dead cells around the neurosphere. Scale bar, 500 μm.

after 48 h with CellTiter-Glo reagent (Promega, Madison, WI, USA) according to the manufacturer's instructions. Briefly, 50 μL of CellTiter-Glo was added to wells containing 50 μL of medium. The plates were shaken for 2 min and incubated for 20 min at room temperature, and luminescence was determined on a Wallac 1420 ARVOsx (PerkinElmer, Norwalk, CT, USA).

## Time-course cytotoxicity test using a microsphere array (MSA)

The MSA (MSE24-CA300, 652 microwells/array; STEM Biomethod, Fukuoka, Japan) was set into one well of a 24-well plate. The neurospheres were dissociated into single cells and seeded into the MSA at a density of 200 cells/microwell (day −1). On day 0, the pretreatment state was recorded by phase-contrast images of neurospheres in the MSA microwells (Figs. 1A and 1B), and etoposide (0, 0.1, 1, and 10 μM) or cisplatin, mercaptopurine, or methotrexate (0, 1, 10, and 100 μM) was applied (control, low, middle, and high concentrations, respectively). This medium was not replaced during the 7-day experiment. Phase-contrast images were captured on days 1, 2, 3, and 7 to monitor neurospheres in the microwells (Figs. 1A and 1B), and the projected areas of neurospheres (*Mori et al., 2006*) were measured using a pen tablet (Intuos, CTH-480; Wacom, Saitama,

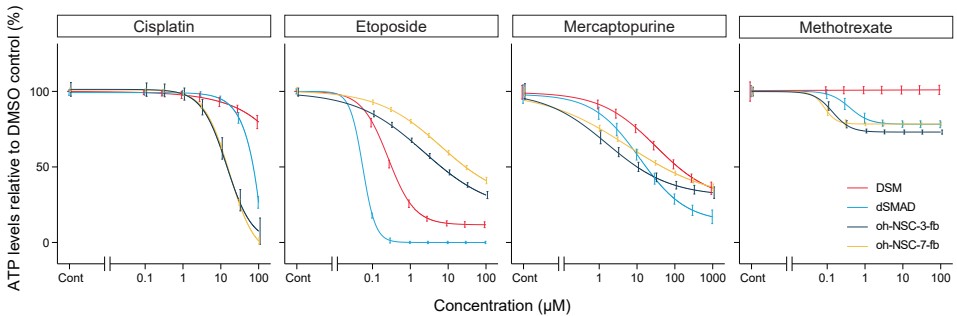

**Figure 2** **Dose-response curves and IC$_{50}$ values obtained from a conventional ATP assay on day 2. control (%).** The $x$-axis indicates the drug concentration ($\mu$M) in log scale, and the $y$-axis indicates ATP levels in the treated cells relative to the DMSO control (%). Colors indicate cell type. The log-logistic model (LL2.4) was used. Error bars represent the 95% CI.

Japan) with the TrakEM2 plugin (*Cardona et al., 2012*) in ImageJ (Fiji package) (*Schindelin et al., 2012*; *Schindelin et al., 2015*). Microwells containing multiple neurospheres on day 0 were excluded from analysis. Outliers in a boxplot of neurosphere sizes on day 0 were also excluded. At each time-point, the neurosphere size in the presence of each treatment was expressed as a percentage of the day 0 value. These data were then expressed as a percentage of the respective DMSO control.

## Statistical analysis

For the endpoint assay (Fig. 2), predicted dose–response curves and 50% inhibitory concentration (IC$_{50}$) values were obtained using the four-parameter log-logistic (LL2.4) and ED functions, respectively, of the drc package (*Ritz et al., 2015*) in R (*R Core Team, 2015*). For the time-course cytotoxicity test, changes in neurosphere size relative to those in the DMSO controls (Fig. 3) and day 0 neurospheres (Fig. S1) were analyzed with the four-parameter logistic model (L.4) and the Brain–Cousens five-parameter model (BC.5) of the drc package in R, respectively. Data from the endpoint assay (Fig. 2) and time-course cytotoxicity test (Figs. 3 and S1) were plotted with 95% confidence intervals (95% CI).

## RESULTS

The IC$_{50}$ of cisplatin, etoposide, mercaptopurine, and methotrexate in the two hiPSC-NSPC cell lines (DSM and dSMAD) and the two hN-NSPC cell lines (oh-NSC-3-fb and oh-NSC-7-fb; Fig. 2 and Table 1) was determined by endpoint (ATP) assay. Cisplatin and etoposide were preferentially toxic to hN-NSPCs and hiPSC-NSPCs, respectively. Interestingly, the two hiPSC-NSPC lines differed in sensitivity to these drugs (Table 1). Mercaptopurine was highly toxic to both types of NSPCs, while methotrexate had almost no effect on either type even at high concentrations. Of the two hiPSC-NSPC lines, DSM was significantly more resistant to the drugs.

To follow the real-time effects of these drugs after treatment, we conducted a simple time-course cytotoxicity test using an MSA (Figs. 1A and 1B). Similar to the ATP assay,

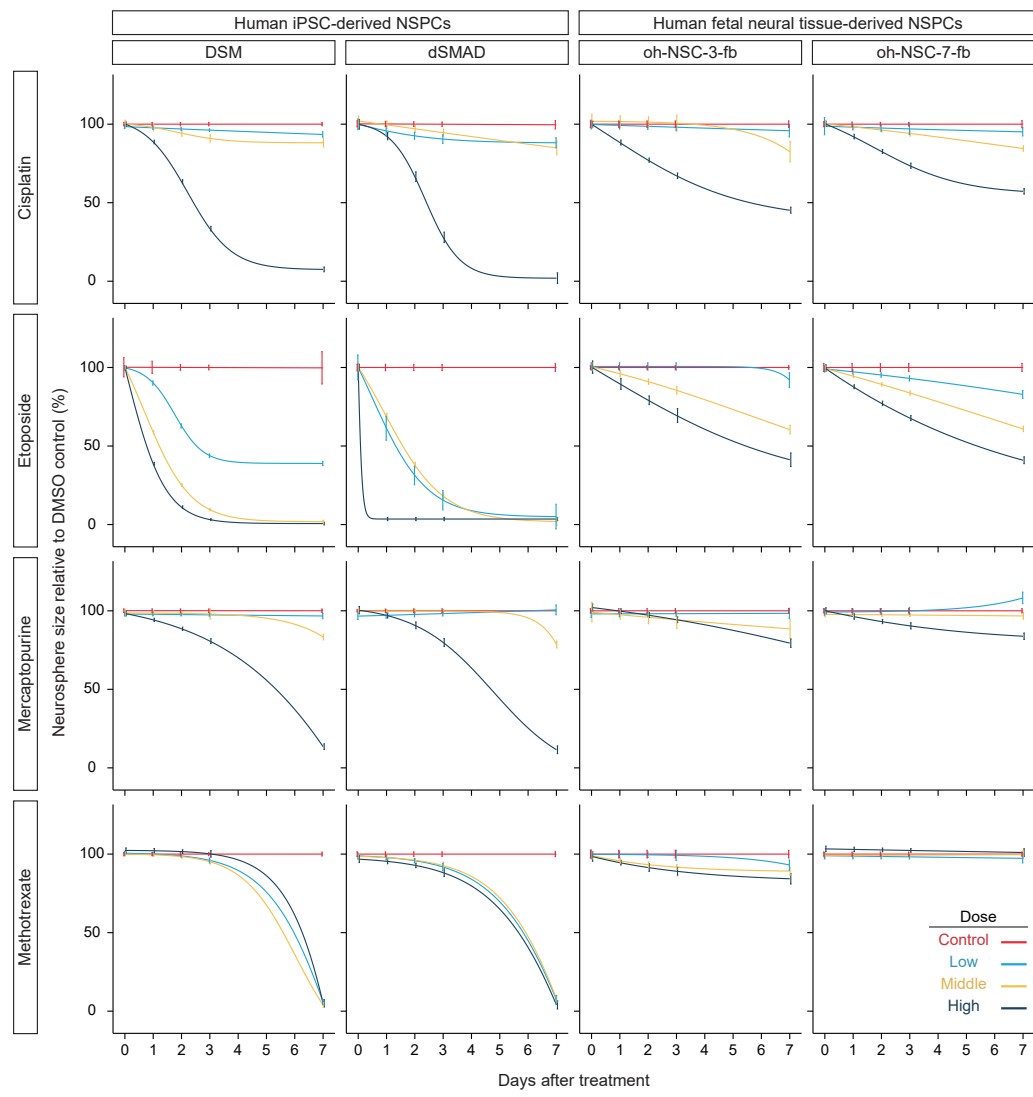

**Figure 3** **Results of time-course cytotoxicity test.** The *x*-axis indicates days after treatment, and the y-axis indicates neurosphere size relative to the DMSO control (%). Colors indicate drug concentrations. The logistic model (L.4) was used. Error bars represent the 95% CI.

**Table 1** IC$_{50}$ values ($\mu$M) of drugs against hiPSC-NSPCs and hN-NSPCs.

| Drug | hiPSC-NSPCs | | hN-NSPCs | |
|---|---|---|---|---|
| | **DSM** | **dSMAD** | **oh-NSC-3-fb** | **oh-NSC-7-fb** |
| Cisplatin | 100< | 72.3 | 14.6 | 15.2 |
| Etoposide | 0.32 | 0.04 | 6.59 | 28.5 |
| Mercaptopurine | 120 | 17.3 | 17.3 | 47.0 |
| Methotrexate | 100< | 100< | 100< | 100< |

we assessed the effect of cisplatin, etoposide, mercaptopurine, and methotrexate at four dosage levels in the two hiPSC-NSPC and two hN-NSPCs lines, in this case by measuring neurosphere size on each day for seven days (Fig. 3 and Table S1). Because neurosphere size of the DMSO control increased during the 7-day assay (Fig. S1), smaller neurosphere sizes of the drug treatment group than those of the DMSO control indicate cytotoxicity of drugs.

Compared to the DMSO control, cisplatin showed toxic effects in hiPSC-NSPCs at low to high concentrations and in hN-NSPCs at middle and high concentrations (Fig. 3). Notably, high concentrations of cisplatin killed hiPSC-NSPC neurospheres, as indicated by their disappearance (Fig. S1 shows changes in neurosphere size relative to day 0), whereas hN-NSPC neurospheres were present throughout the assay period (Fig. S1). Etoposide also affected the neurosphere size in both hiPSC-NSPCs and hN-NSPCs, but it showed earlier and stronger toxicity at lower concentrations compared to cisplatin (Fig. 3). However, the hN-NSPC neurosphere size remained constant even at the highest concentration of etoposide or cisplatin (Fig. S1). Thus, there was a clear difference between hiPSC-NSPCs and hN-NSPCs at the highest concentrations of both cisplatin and etoposide. Moreover, low concentrations of etoposide affected the neurosphere size in dSMAD and oh-NSC-7-fb cells more strongly than that in DSM or oh-NSC-3-fb cells, respectively. Thus, the preferential toxicity of the drug differed not only between the two types of NSPCs, but also between two cell lines of the same type.

Mercaptopurine and methotrexate, even at the highest concentrations, were only mildly toxic to hN-NSPCs and did not stop their growth (Fig. S1). However, hiPSC-NSPCs were affected by both mercaptopurine and methotrexate. At low concentrations, methotrexate was noticeably more toxic to hiPSC-NSPCs than was mercaptopurine (Fig. 3). Unlike cisplatin and etoposide, mercaptopurine and methotrexate had only a limited effect at even the highest concentrations until day 3, which allowed the growth of larger neurospheres (Fig. S1). As with cisplatin and etoposide, mercaptopurine and methotrexate proved to be highly cytotoxic by the end of the assay. From these results, cisplatin and etoposide can be classified as fast-acting drugs with early cellular toxicity, while mercaptopurine and methotrexate can be classified as slow-acting drugs with later toxicity.

## DISCUSSION

Although hiPSC-NSPC transplantation is effective for treating spinal cord injury and stroke, the use of stem cells poses certain risks due to their intrinsic proliferative potential. In particular, the artificial generation of hiPSCs may cause genetic and epigenetic abnormalities, which could potentially increase the risk of tumorigenesis (*Nagoshi & Okano, 2017*). These risks can be reduced prior to transplantation by inserting a 'suicide gene' into the donor cells, or by pretreating the donor cells with inhibitors that reduce their stemness and direct differentiation. However, neither of these strategies is ideal. Transgenes can potentially introduce new risks via genome modification. Although pretreatment can reduce donor-cell stemness, we have not yet found a way to deal with the overgrowth of grafted cells after transplantation. Thus, as a backup safety measure for stem-cell therapies, it is essential to identify drugs that act specifically on the grafted cells, but not on resident

stem cells. To this end, we conducted a small-scale screening of four anticancer drugs and examined their effect on hiPSC-NSPCs and on hN-NSPCs, which are considered to mimic the resident stem cells in the host body.

Based on ATP assay results, we classified the four drugs as follows: (1) more toxic to hN-NSPCs than hiPSC-NSPCs (cisplatin), (2) more toxic to hiPSC-NSPCs than hN-NSPCs (etoposide), (3) similar toxic effects on hiPSC-NSPCs and hN-NSPCs (mercaptopurine), and (4) almost no effect on hiPSC-NSPCs or hN-NSPCs (methotrexate). These results identified etoposide as a candidate backup safety measure for stem cell-based therapies, since it was selectively toxic to hiPSC-NSPCs. However, the dose may need to be adjusted for individual cell lines, since different lines of the same type of NSPC differed in sensitivity to etoposide. Although an ATP assay is useful for characterizing drugs based on dose-response curves and $IC_{50}$ values, the long-term monitoring of transplanted cells is necessary after treatment *in vivo*.

In this study, we monitored the effects of anticancer drugs on neurosphere size *in vitro* for a period of seven days after treatment. Based on these results, we classified cisplatin and etoposide as fast-acting drugs with early cytotoxicity, and mercaptopurine and methotrexate as slow-acting drugs with late cytotoxicity; this classification is consistent with the drug categories. Cisplatin and etoposide are cytotoxic drugs that act directly by alkylating DNA and inhibiting topoisomerase, respectively, whereas mercaptopurine and methotrexate are cytostatic drugs that inhibit IMP dehydrogenase and dihydrofolate reductase, respectively. Compared to hiPSC-NSPCs, the hN-NSPCs were more resistant to high concentrations of cisplatin or etoposide; this difference might be due to the different developmental stages of the cells. In fact, hiPSC-NSPCs recapitulate regular neural development along with cell proliferation after transplantation (*Sugai et al., 2016*), and this characteristic will likely be useful for developing drugs that specifically target transplanted cells. The presence of mercaptopurine and methotrexate, which are cytostatic, eventually induced death in hiPSC-NSPCs, but only mildly limited hN-NSPC growth during the 7-day assay. Mercaptopurine and the cytotoxic drugs cisplatin and etoposide decreased the ATP level in both hN-NSPCs and hiPSC-NSPCs on day 2 of treatment (Fig. 2). Although ATP level is a useful index of cell viability, mercaptopurine-mediated inhibition of *de novo* purine synthesis might also reduce the ATP level in the absence of cell death, in contrast to other cytotoxic drugs. However, the effect of cytostatic drugs distinguished hN-NSPCs and hiPSC-NSPCs in the time-course assay (Fig. 3). Therefore, we need to reassess cytostatic drugs from the viewpoint of their time-dependent action. Our findings indicate that methotrexate is preferable to mercaptopurine as a candidate safety measure for hiPSC-NSPC transplantation because it was cytotoxic even at low concentrations.

This study has certain limitations. First, the mechanisms underlying the late toxicity of cytostatic drugs on hiPSC-NSPCs are unknown, although hN-NSPCs are reported to express high levels of ABCB1 transporter (*Islam et al., 2005*; *Yamamoto et al., 2009*), which may contribute to a development of tolerance to slow-acting drugs. Further studies are needed to elucidate the mechanisms underlying the selective cytotoxic effects of cytostatic

drugs on hiPSC-NSPCs. Second, these effects were obtained *in vitro* and therefore await confirmation *in vivo*.

## CONCLUSION

Based on a 7-day time-course cytotoxicity test, we classified four anticancer drugs as fast-acting or slow-acting. We found that the slow-acting drugs affected hiPSC-NSPCs and hN-NSPCs differently, which was not evident in a conventional ATP assay performed on day 2. As hN-NSPCs were more tolerant of slow-acting drugs than hiPSC-NSPCs, we propose that slow-acting drugs such as methotrexate may provide drug candidates for backup safety measures to prevent the undesirable proliferation of hiPSC-NSPCs after transplantation therapies.

## ACKNOWLEDGEMENTS

The authors thank Ms. Ai Takada, Ms. Miho Sumida, Ms. Ema Yoshioka, Ms. Yui Inazawa, and Mr. Daisuke Kanematsu for technical support.

### Funding

This study was supported by the Research on Regulatory Harmonization and Evaluation of Pharmaceuticals, Medical Devices, Regenerative and Cellular Therapy Products, Gene Therapy Products, and Cosmetics from the Japan Agency for Medical Research and Development (AMED), and the Advanced Research for Medical Products Mining Programme of the National Institute of Biomedical Innovation (NIBIO). There was no additional external funding received for this study. The funders had no role in study design, data collection and analysis, decision to publish, or preparation of the manuscript.

### Grant Disclosures

The following grant information was disclosed by the authors:
Japan Agency for Medical Research and Development (AMED).
National Institute of Biomedical Innovation (NIBIO).

### Competing Interests

The authors declare there are no competing interests.

### Author Contributions

- Hayato Fukusumi conceived and designed the experiments, performed the experiments, analyzed the data, contributed reagents/materials/analysis tools, wrote the paper, prepared figures and/or tables, reviewed drafts of the paper.
- Yukako Handa performed the experiments, analyzed the data, contributed reagents/materials/analysis tools, prepared figures and/or tables, reviewed drafts of the paper.
- Tomoko Shofuda conceived and designed the experiments, performed the experiments, contributed reagents/materials/analysis tools, reviewed drafts of the paper.
- Yonehiro Kanemura conceived and designed the experiments, wrote the paper, reviewed drafts of the paper.

### Ethics

The following information was supplied relating to ethical approvals (i.e., approving body and any reference numbers):

The use of hN-NSPCs and hiPSCs was approved by the Osaka National Hospital hN-NSPCs and hiPSCs ethics committee (Nos. 110, 120, and 146).

### Data Availability

The raw data is included in Tables S2 and S3.

### Supplemental Information

Supplemental information for this article can be found online at http://dx.doi.org/10.7717/peerj.4187#supplemental-information.

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
