# Peer review of "Small-scale screening of anticancer drugs acting specifically on neural stem/progenitor cells derived from human-induced pluripotent stem cells using a time-course cytotoxicity test"

_PeerJ, doi:10.7717/peerj.4187_

## Round 0.1 · original submission · Minor Revisions

Dear Authors,

Thank you for submitting your manuscript to PeerJ. Following the review of Research Article titled "Small-scale screening of anticancer drugs acting specifically on neural stem/progenitor cells derived from human induced pluripotent stem cells using a time-course cytotoxicity test", I recommend that it should be revised taking into account the changes requested by the reviewers.

Best regards

Reviewer 1 ·

Basic reporting

no comment

Experimental design

Well-designed

Validity of the findings

The finding that the slow-acting drug, methotrexate has cytotoxic effects on human iPS cell-derived NSPCs specifically but not on human tissue-derived NSPCs is new and interesting.

Additional comments

This article provides very valuable information for the development of drug discovery toxicology, and could be published if the authors modify the text as directed.

1, Time-course assay was performed by image-based and noninvasive procedure. I think that it is better for the authors to emphasize this a little more. Because the acceptance of image-based morphological assays of cells and living matters has been increased with advances in information technology currently.

2, Authors should describe a relational expression to calculate the percentage of neurosphere size indicated in y-axis of Fig. 3.

3, Authors should indicate the number of samples used in the experiment.

Reviewer 2 ·

Basic reporting

The paper is well organized.

Experimental design

1. In the time-course assay, the drug responses of cells may be varied by the neurosphere sizes. Why did the authors use the cell density of 200 cells/micowell?

2. Did the authors exchange the culture medium including drug in the time-course assay?

Validity of the findings

This study describes the efficiency of time-course cytotoxicity assay using 3D (neurosphere) culture. The idea that this assay applies as a backup safety measure for stem-cell therapies is unique and original. Furthermore, this assay has potential of wide use for drug screening method, because the protocol is very simple.

The findings are novel and consistent. However, the following point is unclear.

It is difficult to understand the degree of cytotoxic effect in the time-course assay, compared to the endpoint assay. Please add the discussion about the criteria for judgment of cytotoxicity on the time-course assay.

·

Basic reporting

No comments

Experimental design

The authors wisely combined two methods, one using the ATP measurement for cell proliferation/viable cells and cytotoxic effects of drugs after 2 days and the second (MSA assay) to follow the real time effects of the drugs by measuring the neurosphere size during 7 days. This expermimental design was relevant and meaningful.

Validity of the findings

The combination of these two methods allowed to refine the effects of drugs concentration and with time. These results allowed to classify cisplatin and etoposide as fast-acting drugs with early cytotoxicity, and mercaptopurine and methotrexate as slow-acting drugs with late cytotoxicity. These results highlighted the need to reassess cytostatic drugs from the viewpoint of their time-dependent action. These effects were obtained in vitro and should be confirmed in vivo.

Additional comments

This manuscript approaches scientifically how to overcome the risks of teratoma formation and the overgrowth of transplanted cells. This manuscript describes concicely but very convingly and elegantly the specificity of drugs able to reduce the potential tumorigenicity of transplanted cells.
The paper is well written with figures highly informative.

·

Basic reporting

The article is well written, clear description of findings.
The topic is very important,

Experimental design

The experimental design is good, although the work would benefit from inclusion more compounds into studies

Validity of the findings

The findings are valid, but somewhat limited

Additional comments

The recommendation would be to extend study to include more compounds. Also, the paper would benefit from more discussion of challenges of cell therapies and guidance for the characterization of stem cell tissues for transplantation. For example, recent works of S. Yamanaka were related to characterization of proliferation potential of stem cell derived tissues and also genotyping of cell lines used for transplantation. More discussion may be beneficial for the paper.

---

## Round 0.2 · accepted · Accept

Dear Yonehiro,

Thank you for your submission to PeerJ - your article is now Accepted.